# How Normalization and Weight Decay Can Affect SGD? Insights from a Simple Normalized Model

## Abstract

Recent works (Li et al., 2020; Wan et al., 2021) characterize an important mechanism of normalized model trained with SGD and WD (Weight Decay), called Spherical Motion Dynamics (SMD), confirming its widespread effects in practice. However, no theoretical study is available on the influence of SMD on the evolution of the loss of normalized models in literature. In this work, we seek to understand the effect of SMD by theoretically analyzing a simple normalized model, named as Noisy Rayleigh Quotient (NRQ). On NRQ, We theoretically prove SMD can dominate the whole training process via controlling the evolution of angular update (AU), an essential feature of SMD. Specifically, we show: 1) within equilibrium state of SMD, the convergence rate and limiting risk of NRQ are mainly determined by the theoretical value of AU; and 2) beyond equilibrium state, the evolution of AU can interfere the optimization trajectory, causing odd phenomena such as "escape" behavior. We further show the insights drawn from NRQ is consistent with empirical observations in experiments on real datasets. We believe our theoretical results shed new light on the role of normalization techniques during the training of modern deep learning models.

## 1 Introduction

Normalization (Ioffe & Szegedy, 2015; Wu & He, 2018) is one of the most widely used deep learning techniques, and has become an indispensable part in almost all popular architectures of deep neural networks. Though the success of normalization techniques is indubitable, its underlying mechanism still remains mysterious, and has become a hot topic in the realm of deep learning.

Many works have contributed in figuring out the mechanism of normalization from different aspects. While some works (Ioffe & Szegedy, 2015; Santurkar et al., 2018; Hoffer et al., 2018; Bjorck et al., 2018; Summers & Dinneen, 2019; De & Smith, 2020) focus on intuitive reasoning or empirical study, others (Dukler et al., 2020; Kohler et al., 2019; Cai et al., 2019; Arora et al., 2018; Yang et al., 2018; Wu et al., 2020) focus on establishing theoretical foundation. A series of works (Van Laarhoven, 2017; Chiley et al., 2019; Kunin et al., 2021; Li et al., 2020; Wan et al., 2021; Lobacheva et al., 2021; Li & Arora, 2019) have noted that, in practical implementation, the gradient of normalized models is usually computed in a straightforward manner which results in its scale-invariant property during training. The gradient of a scale-invariant weight is always orthogonal to the weight, and thus makes the training trajectory behave as motion on a sphere. Besides, in practice, many models are trained using SGD with Weight Decay (WD), hence normalization and WD in SGD can cause a so-called "equilibrium" state, in which the effect of gradient and WD on weight norm cancel out (see Fig. 1(a)).

It has been a long time since the concept of equilibrium was first proposed (Van Laarhoven, 2017) while either theoretical justification or experimental evidence had still been lacking until recently. Recent works (Li et al., 2020; Wan et al., 2021) theoretically justify the existence of equilibrium in both theoretical and empirical aspects, and characterize the underlying mechanism that yields equilibrium, named as "Spherical Motion Dynamics". In Wan et al. (2021) the authors further show SMD exists in a wide range of computer vision tasks, including ImageNet Deng et al. (2009) and MSCOCO (Lin et al., 2014). More detailed review can be seen in appendix A.

Though the existence of SMD has been confirmed both theoretically and empirically, as well as some of its characteristics, we notice that so far no previous work has ever theoretically justified how SMD can affect the evolution of the loss of normalized models. Although some attempts have been made in Li et al. (2020); Wan et al. (2021) to explore the role of SMD in the training by conjectures and empirical studies, they still lack theoretical justification on their findings. In hindsight, the main challenge to theoretically analyze the effect of SMD is that SMD comes from the joint effect of normalization and WD

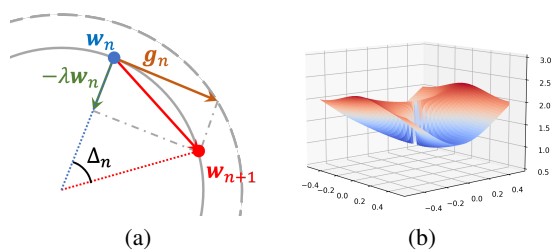

(a)            (b)

Figure 1: (a) Illustration of Spherical Motion Dynamics; (b) Loss landscape of a Rayleigh Quotient with WD ($l^2$ regularization): $x^2 + 2y^2/(x^2+y^2) + (x^2+y^2)$

which can significantly distort the loss landscape (see Figure 1(b)), and thus dramatically weaken some commonly used assumptions such as (locally) convexity, Lipschitz continuity, etc. Exploring the optimization trajectory on such distorted loss landscape is very challenging, much less that taking in addition SMD into account in the consideration.

In this paper, as the first significant attempt to overcome the challenge on studying the effect of SMD towards evolution of the loss, we propose a simple yet representative normalized model, and theoretically analyze how SMD influences the optimization trajectory. We adopt the SDE framework of Li et al. (2020) to derive the analytical results on the evolution of NRQ, and concepts of Wan et al. (2021) to interpret the theoretical results we obtain in this paper. Our contributions are

- We design a simple normalized model, named as Noisy Rayleigh Quotient (NRQ). NRQ possesses all the necessary properties to induce SMD, consistent with those in real neural networks. NRQ contributes a powerful tool for analyzing how normalization affects first-order optimization algorithms;

- We derive the analytical results on the limiting dynamics and the stationary distribution of NRQ. Our results show the influence of SMD is mainly reflected on how angular update (AU), a crucial feature of SMD, affects the convergence rate and limiting risk. We discuss the influence of AU within equilibrium and beyond equilibrium respectively, figuring out the association between the evolution of AU and the evolution of the optimization trajectory in NRQ;

- We show that the insights drawn from the theoretical results on NRQ can adequately interpret typical observations in deep learning experiments. Specifically, we confirm the role of learning rate and WD is equivalent to that of scale-invariant weight in SGD. We show the Gaussian type initialization strategy can affect the training process only because it can change the evolution of AU at the beginning. We also confirm that under certain condition, SMD may induce "escape" behavior of optimization trajectory, resulting in "pseudo overfitting" phenomenon in practice.

## 2   NOISY RAYLEIGH QUOTIENT

### 2.1   PROBLEM SET UP

We use Rayleigh Quotient Horn & Johnson (2012) as the objective function, defined as

$$\mathcal{L}(\boldsymbol{X}) = \frac{\boldsymbol{X}^T \boldsymbol{A} \boldsymbol{X}}{2 \boldsymbol{X}^T \boldsymbol{X}}, \tag{1}$$

where $\boldsymbol{X} \in \mathbb{R}^p \backslash \{\boldsymbol{0}\}$, $\boldsymbol{A} \in \mathbb{R}^{p \times p}$ is positive semi-definite. Based on its form, Rayleigh Quotient is equivalent to a quadratic function using weight normalization (Salimans & Kingma, 2016).

Now considering the following optimization problem

$$\min_{\boldsymbol{X} \in \mathbb{R}^p \backslash \{\boldsymbol{0}\}} \mathcal{L}(\boldsymbol{X}), \tag{2}$$

Obviously the solutions to equation 2 are $\mathcal{X}^* = \{\alpha\boldsymbol{v}|\alpha \in \mathbb{R}^+, \boldsymbol{v} \in \mathcal{S}^{p-1}, \boldsymbol{A}\boldsymbol{v} = \lambda_1\boldsymbol{v}, \lambda_1 \text{ is the smallest eigen value of } \boldsymbol{A}\}$. Consider solving Eq equation 2 by Stochastic Gradient Descent (SGD) with constant learning rate (LR) $\eta > 0$ and **Weight Decay (WD)**, the update rule at step $n$ is

$$\boldsymbol{X}_{n+1} = \boldsymbol{X}_n - \eta\boldsymbol{g}_n - \lambda\eta\boldsymbol{X}_n, \tag{3}$$

where $-\lambda\eta\boldsymbol{X}_n$ is the weight decay part with a positive factor $\lambda$; $\boldsymbol{g}_n$ is the stochastic gradient of equation 1 at step $n$. Inspired by Zhang et al. (2019), the gradient noise is constructed as Gaussian noise to simulate the "mini-batch training". Specifically, $\boldsymbol{g}_n$ is constructed as

$$\boldsymbol{g}_n = \frac{1}{||\boldsymbol{X}_n||_2}\boldsymbol{P}_n\boldsymbol{A}\tilde{\boldsymbol{X}}_n + \frac{1}{||\boldsymbol{X}_n||_2}\boldsymbol{P}_n(\tilde{\boldsymbol{\Sigma}})^{1/2}\boldsymbol{\varepsilon}_n, \tag{4}$$

where $\tilde{\boldsymbol{X}}_n \triangleq \boldsymbol{X}_n/||\boldsymbol{X}_n||_2$; $\boldsymbol{P}_n \triangleq (\boldsymbol{I} - \tilde{\boldsymbol{X}}_n\tilde{\boldsymbol{X}}_n^T)$; $\tilde{\boldsymbol{\Sigma}} \in \mathbb{R}^{p \times p}$ is a positive definite matrix; $\boldsymbol{\varepsilon}_n \sim \mathcal{N}(\boldsymbol{0}, \boldsymbol{I})$. Then we have

$$\mathbb{E}\boldsymbol{g}_n = \nabla_{\boldsymbol{X}}\mathcal{L}(\boldsymbol{X}_n), \quad Cov(\boldsymbol{g}_n) = \frac{1}{||\boldsymbol{X}||_2^2}\boldsymbol{P}_n(\tilde{\boldsymbol{\Sigma}})\boldsymbol{P}_n. \tag{5}$$

$\boldsymbol{g}_n$ defined as equation 4 can simulate the mini-batch stochastic gradient of a scale-invariant loss because

$$\langle\boldsymbol{g}_n, \boldsymbol{X}_n\rangle = 0, \quad \boldsymbol{g}_n = \frac{1}{k}\boldsymbol{g}_n\big|_{\boldsymbol{X}=k\boldsymbol{X}_n}, \quad \forall k > 0, \tag{6}$$

which are necessary conditions to let SMD occur during the evolution of SGD (Li et al., 2020; Wan et al., 2021). We call the process *Noisy Rayleigh Quotient (NRQ)* which optimizes the Rayleigh Quotient equation 3 using stochastic gradient Eq equation 4 as .

*Remark* 1. The form of stochastic gradient of NRQ can be regarded as the normalized form of Noisy Quadratic Model (NQM) (Zhang et al., 2019), in which the objective function is

$$\mathcal{L}(\boldsymbol{X}) = \frac{1}{2}\boldsymbol{X}^T\boldsymbol{A}\boldsymbol{X} \tag{7}$$

and the stochastic gradient is defined as

$$\boldsymbol{g}_n = \boldsymbol{A}\boldsymbol{X}_n + \boldsymbol{\Sigma}^{1/2}\boldsymbol{\varepsilon}_n. \tag{8}$$

But the dynamics of NQM and NRQ are quite different: NQM is basically a convex model and has only one optimal solution $\boldsymbol{0}$, while NRQ is a nonconvex problem and has an infinite number of solutions ($\mathcal{X}^*$), thus making it much more difficult to analyze comparing with NQM.

## 2.2 APPROXIMATE SGD AS STOCHASTIC DIFFERENTIAL EQUATION

Though the thorough analysis on the characteristics of SMD is established on the discrete form (Wan et al., 2021), directly analyzing evolution dynamics of equation 3 taking SMD into account in discrete form is still too complex. On the other hand, we can tackle the problem using the SDE approximation introduced in Li et al. (2020), which has established the continuous form of SMD.

Using SDE approximation, the evolution dynamics of equation 3 can be approximated by

$$d\boldsymbol{X}_t = -\eta\Big(\frac{1}{||\boldsymbol{X}_t||_2}\boldsymbol{P}_t\boldsymbol{A}\tilde{\boldsymbol{X}}_t + \lambda\boldsymbol{X}_t\Big)dt + \frac{\eta\boldsymbol{P}_t(\tilde{\boldsymbol{\Sigma}})^{1/2}}{||\boldsymbol{X}_t||_2}d\boldsymbol{B}_t, \tag{9}$$

where $\boldsymbol{B}_t$ is a $p$-dimensional Brownian motion. Here we follow the form of SDE used in Li et al. (2020) instead of the commonly used form proposed in Li et al. (2017) by extracting a LR factor $\eta$ from the differential time $dt$. The extracted factor is useful in connecting the characteristics of SMD in discrete setting and continuous setting.

Due to the scale-invariant property, $||\boldsymbol{X}_t||_2$ cannot influence the Rayleigh Quotient $\mathcal{L}(\boldsymbol{X}_t)$ at all, the intrinsic domain of NRQ is a unit sphere $\mathcal{S}^{p-1}$ (Li et al., 2020). But $||\boldsymbol{X}_t||_2$ may affect the evolution dynamics of NRQ since it is involved in the system equation 9. To decouple the evolution of $\boldsymbol{X}_t$ on its intrinsic domain (i.e. the evolution of $\tilde{\boldsymbol{X}}_t$), and the evolution of $||\boldsymbol{X}_t||_2$, according to Li et al. (2020), let $M_t \triangleq ||\boldsymbol{X}_t||_2$, then equation 9 can be rewritten as the following two SDEs:

$$d\tilde{\boldsymbol{X}}_t = -\Big[\frac{\eta}{M_t}\boldsymbol{P}_t\boldsymbol{A}\tilde{\boldsymbol{X}}_t + \frac{\eta^2}{2M_t^2}\text{Tr}(\boldsymbol{P}_t\tilde{\boldsymbol{\Sigma}}\boldsymbol{P}_t)\tilde{\boldsymbol{X}}_t\Big]dt - \frac{\eta}{M_t}\boldsymbol{P}_t\tilde{\boldsymbol{\Sigma}}^{\frac{1}{2}}d\boldsymbol{B}_t \tag{10}$$

$$dM_t = \Big[-2\lambda\eta M_t + \frac{\eta^2}{M_t}\text{Tr}(\boldsymbol{P}_t\tilde{\boldsymbol{\Sigma}})\Big]dt \tag{11}$$

Note the diffusion part is missing in Eq equation 11, so it is possible to derive the explicit solution of Eq equation 11, computed as

$$M_t^2 = e^{-4\lambda\eta t}M_0^2 + 2\eta^2 \int_0^t e^{-4\lambda\eta(t-\tau)}\operatorname{Tr}(\boldsymbol{P}_\tau\tilde{\boldsymbol{\Sigma}})d\tau. \tag{12}$$

## 2.3 CHARACTERISTICS OF SMD IN NRQ

Previous works (Van Laarhoven, 2017; Chiley et al., 2019; Kunin et al., 2021; Li et al., 2020) usually regard the convergence of weight norm as the sign of equilibrium state in SMD. However, Wan et al. (2021) argues that equilibrium of SMD in practice is actually a dynamic state, where the convergence of weight norm may not hold when the variance of gradient noise on intrinsic domain varies dramatically during the whole training process. Notwithstanding, Wan et al. (2021) reveals another essential characteristic of equilibrium regardless of the convergence of the norm: the AU, defined as $\Delta_n = \angle(\boldsymbol{X}_n, \boldsymbol{X}_{n+1})$. When equilibrium of SMD is reached, AU will satisfy $\mathbb{E}\Delta_n = \sqrt{2\lambda\eta}$. In NRQ, the (discrete) AU can be computed by

$$\Delta_n = \angle(\boldsymbol{X}_n, \boldsymbol{X}_{n+1}) = \arctan(\frac{||\boldsymbol{g}_n||\eta}{||\boldsymbol{X}_n||_2}) \approx \frac{||\boldsymbol{g}_n||\eta}{||\boldsymbol{X}_n||_2}. \tag{13}$$

Then we have

$$\mathbb{E}\Delta_n^2 = \frac{[||\nabla_{\boldsymbol{X}}\mathcal{L}(\tilde{\boldsymbol{X}}_n)||_2^2 + \operatorname{Tr}(\boldsymbol{P}_n\tilde{\boldsymbol{\Sigma}})]\eta^2}{||\boldsymbol{X}_n||_2^4}. \tag{14}$$

Though AU has specific geometric meaning in discrete form, as it represents the geodesic distance between $\tilde{\boldsymbol{X}}_n$ and $\tilde{\boldsymbol{X}}_{n+1}$ on unit sphere $\mathcal{S}^{p-1}$, its definition cannot be applied directly in continuous setting. To connect the concept of SMD in discrete setting and continuous setting, a continuous version of AU in NRQ is defined as

**Definition 1** (AU). $\mathbb{E}||\tilde{\boldsymbol{X}}_\tau - \tilde{\boldsymbol{X}}_t||_2^2$ is differentiable on $[t, +\infty)$, then AU at $t$ is defined as

$$\Delta_t = \sqrt{\lim_{\tau\to t}\frac{\mathbb{E}_t||\tilde{\boldsymbol{X}}_\tau - \tilde{\boldsymbol{X}}_t||_2^2}{\tau - t}}. \tag{15}$$

*Remark* 2. The definition of AU in continuous setting is inspired from the concept of "angular velocity" used in Kunin et al. (2021), in which the author formulated the equilibrium of SMD using gradient flow.

This definition relies on the fact that $\mathbb{E}||\tilde{\boldsymbol{X}}_\tau - \tilde{\boldsymbol{X}}_t||_2^2$ is differentiable on $t$. By the definition we can derive the following properties of AU in NRQ:

**Lemma 1.** *In the evolution of equation 10, equation 11, we have* $\Delta_t^2 = \frac{\operatorname{Tr}(\boldsymbol{P}_t\tilde{\boldsymbol{\Sigma}})\eta^2}{M_t^2}$. *If* $\operatorname{Tr}(\boldsymbol{P}_t\tilde{\boldsymbol{\Sigma}})$ *is constant, then* $\lim_{t\to\infty}\Delta_t = \sqrt{2\lambda\eta}$.

The proof is in Appendix B.1. Comparing with equation 14 and lemma 1, the theoretical value $\mathbb{E}\Delta_n^2$ in discrete form is similar to its continuous form except for an additional term $||\nabla_{\boldsymbol{X}}\mathcal{L}(\tilde{\boldsymbol{X}}_n)||_2^2$ in the top of fraction, denoting the full gradient norm. Note when $\tilde{\boldsymbol{X}}_n$ is close to its optimal point, this term is usually close to zero, hence can be ignored comparing with the magnitude of gradient noise $\operatorname{Tr}(\boldsymbol{P}_n\boldsymbol{\Sigma}_n)$. This is called noisy dominated regime (Zhang et al., 2019; Smith et al., 2020), which commonly holds in practice especially in large-scale dataset tasks (Smith et al., 2020; Wan et al., 2021), and happen to be the case where discretization error can be controlled (Li et al., 2021). Besides, the limit of $\Delta_t$ is $\sqrt{2\lambda\eta}$, exactly same as the theoretical value of AU in discrete form.

In summary, SMD in continuous form is fundamentally equivalent to SMD in the discrete form in noisy dominated regime, where they share the same characteristics on AU. In the following context, we adopt the unifying concept of SMD and AU, without distinguishing the discrete and continuous form.

## 3 ANALYTICAL RESULTS ON EVOLUTION OF NRQ

First of all, the following two assumptions are introduced to simplify the derivation and highlight the insights of the analytical results:

**Assumption 1.** $A$ is diagonal matrix with diagonal elements in ascending order, i.e. $A = \text{diag}(a_1, a_2, \ldots, a_p)$, $a_1 < a_2 \leq a_3, \ldots, \leq a_p$.

**Assumption 2.** $\exists \sigma > 0, \tilde{\Sigma} = \sigma^2 I$.

In assumption 1, $A$ is diagonalized to simplify derivation, same as Zhang et al. (2019) does in the quadratic model. We further assume $a_1 < a_2$ to ensure NRQ has at most 2 solutions $\pm e_1$, where $e_1^{(1)} = 1, e_1^{(i)} = 0, 2 \leq i \leq p - 1$. Assumption 2 is used to ensure $\text{Tr}(P_t \tilde{\Sigma})$ is constant during the whole process. This constant variance of gradient noise assumption are also used in Zhu et al. (2019); Li et al. (2020).

Note even under assumption 1, NRQ still has two different global optimal solutions $\pm e_1$ on $\mathcal{S}^{p-1}$, so directly analyzing the convergence behavior by distance to the optimal points is inappropriate. Therefore, to properly track the optimization trajectory, we analyze the evolution of $f_t \triangleq (e_1^T \tilde{X}_t)^2 = (\tilde{X}_t^{(1)})^2$, where $\tilde{X}_t^{(1)}$ denotes the first element of $\tilde{X}$. Note that $f_t$ is an ideal index to reflect the optimization trajectory because $f_t = 1$ if and only if $X_t = \pm e_1$. Besides, $f_t$ can (roughly) bound the evolution of the loss $L_t$ by

$$a_1 + (a_2 - a_1)(1 - f_t) \leq L_t \leq a_1 + (a_p - a_1)(1 - f_t).$$

Using Itô Lemma, the SDE of $f_t$ can be derived from equation 10 and equation 11, written as

$$df_t = [\frac{\eta(L_t - a_1)}{M_t} f_t + \frac{\eta^2 \sigma^2}{M_t^2}(1 - pf_t)]dt + \frac{2\eta\sigma}{M_t}\sqrt{f_t(1 - f_t)}dB_t, \tag{16}$$

$$M_t^2 = e^{-4\lambda\eta t}(M_0^2 - \frac{(p-1)\sigma^2\eta}{2\lambda}) + \frac{(p-1)\sigma^2\eta}{2\lambda} \tag{17}$$

*Remark* 3. It is possible to directly explore the evolution of the loss $L_t$ by deriving the SDE of $L_t$ using Itô's lemma. But some terms in SDE of $L_t$ is hard to handle comparing with the SDE of $f_t$.

Now we can define the risk of NRQ as $r_t \triangleq 1 - \mathbb{E}f_t$, our first theorem depicts the convergence behavior of NRQ by giving the bounds of the risk;

**Theorem 1.** *The solution of Eqs equation 16 and equation 17 satisfies*

$$r_t \geq e^{-G_1(t)}[1 - f_0 + \int_0^t \Delta_\tau^2 e^{G_1(\tau)}d\tau], \tag{18}$$

*where*

$$G_1(t) \triangleq \int_0^t [\frac{(a_p - a_1)\eta}{\sqrt{p-1}\sigma}\Delta_\tau + \frac{p}{p-1}\Delta_\tau^2]d\tau; \tag{19}$$

*Furthermore, given $\xi \in (0, 1)$, define $\varepsilon(t) = \mathbb{P}(f_t < \xi)$ as the tail probability of $f_t$ dynamics. Then for any $t \geq 0$, we have*

$$r_t \leq e^{-\tilde{G}_1(t)}[1 - f_0 + \int_0^t (\frac{(a_2 - a_1)\eta\xi\varepsilon(t)}{\sqrt{p-1}\sigma}\Delta_\tau + \Delta_\tau^2)e^{-\tilde{G}_\tau}d\tau], \tag{20}$$

*where*

$$\tilde{G}_1(t) \triangleq \int_0^t [\frac{(a_2 - a_1)\eta\xi}{\sqrt{p-1}\sigma}\Delta_\tau + \frac{p}{p-1}\Delta_\tau^2]d\tau; \tag{21}$$

Proof is in Appendix C. Theorem 1 implies the evolution of risk are mostly determined by the evolution of AU. Note the lower bound Eq equation 20 relies on $\xi$ and $\varepsilon(t)$. To make the lower bound tighter, ideally $\xi$ should be close to 1, while $\varepsilon(t)$ should be close to 0. We will discuss $\xi$ and $\varepsilon(t)$ in details later.

### 3.1 EQUILIBRIUM STATE OF SMD

Though an analytical result is shown in Theorem 1, the global picture of the dynamics is still not clear, since the evolution of $r_t$ is associated with the evolution of AU $\Delta_t$. Fortunately, it has been known that equilibrium of SMD must be reached, in which $\Delta_t = \sqrt{2\lambda\eta}$ regardless of the evolution of $\tilde{X}_t$. Hence, we can directly explore the evolution of $r_t$ in equilibrium, as the following corollary shows:

**Corollary 1** (Equilibrium state dynamics). *Assume $M_0 = \sqrt{\frac{\eta(p-1)\sigma^2}{2\lambda}}$, $\lambda\eta \ll 1$, and $\exists\varepsilon > 0$, $\overline{\lim}_{t\to+\infty}\varepsilon(t) \le \varepsilon$ in Theorem 1, then $\exists C > 0$, such that*

$$\underline{r}^* + (1 - f_0 - \underline{r}^*)e^{-\underline{g}_1^* t} \le r_t \le \bar{r}^* + \varepsilon + e^{-\tilde{g}_1^* t}C \tag{22}$$

*where*

$$\underline{g}_1^* = \frac{a_p - a_1}{\sqrt{p-1}\sigma}\sqrt{2\lambda\eta} + \mathcal{O}(\lambda\eta), \quad \tilde{g}_1^* = \frac{\xi(a_p - a_1)}{\sqrt{p-1}\sigma}\sqrt{2\lambda\eta} + \mathcal{O}(\lambda\eta), \tag{23}$$

$$\underline{r}^* = \frac{\sqrt{p-1}\sigma}{a_p - a_1}\sqrt{2\lambda\eta} + \mathcal{O}(\lambda\eta) \quad \bar{r}^* = \frac{\sqrt{p-1}\sigma}{\xi(a_2 - a_1)}\sqrt{2\lambda\eta} + \mathcal{O}(\lambda\eta), \tag{24}$$

Proof is in Appendix D.1. Corollary 1 shows that in equilibrium state of SMD, when $\varepsilon \ll \bar{r}^*$, NRQ still converges in a linear rate regime, where the convergence rate is (roughly) proportional to the AU by equation 23, which is only determined by LR $\eta$ and WD factor $\lambda$. However, the limiting risk is also (roughly) proportional to the AU by equation 24. Thus, a trade-off exists between the convergence rate and limiting risk when tuning AU: large AU can make the loss decrease more quickly at the beginning, but will enlarge the limiting risk in the end, resulting a larger steady loss (see Figure 2 (a)-(d)). This can explain why decreasing LR strategy is always necessary to obtain the best performance when training models in practice.

*Remark* 4. Such trade-off between convergence rate and limiting risk also exists in the convergence behavior of NQM (Zhang et al., 2019). Zhang et al. (2019) claims the trade-off in NQM can be adjusted by tuning LR or gradient noise; while in NRQ, the trade-off can not only be adjusted by LR or gradient noise, but also WD factor. Besides, the association between AU and the convergence rate/limiting risk is consistent with the conjectures about the relation between AU and dynamics of normalized DNN in Wan et al. (2021), in which the authors suppose AU is correlated with the steady loss when training normalized DNN.

**Stationary distribution of $f_t$** Corollary 1 only presents a bound for the risk. With additional assumptions, the stationary distribution of $f_t$ and limiting risk $r_*$ can be explicitly derived using Fokker Planck equation.

**Theorem 2.** *Assume the spectrum of $\mathbf{A}$ takes only 2 distinct real value $a_1 = a_l < a_h = a_2 = \cdots = a_p$. Denote the stationary distribution of $f_t$ by $\rho_*(f)$. Then*

$$\rho_*(f) \propto e^{2\kappa f} f^{-\frac{1}{2}}(1 - f)^{\frac{p-3}{2}}, \ f \in [0, 1] \tag{25}$$

*where $\kappa = \frac{\sqrt{p-1}}{2\sigma\sqrt{2\eta\lambda}}$. In addition, the limit of risk $r_t$ exists and is given by*

$$r_* \triangleq \lim_{t\to\infty} r_t = 1 - \frac{\sqrt{p-1}\sigma}{a_h - a_l}\sqrt{2\lambda\eta} + o(\sqrt{2\lambda\eta}); \tag{26}$$

*Moreover, there exists $\mu, C > 0$, such that for any $\xi \in (0, 1)$, the tail probability $\varepsilon(t) \triangleq \mathbb{P}(f_t < \xi)$ can be governed by*

$$|\varepsilon(t) - \varepsilon_*| \le Ce^{-\mu t} \tag{27}$$

*in which $\varepsilon_*$ is the stationary tail probability $\varepsilon_* \triangleq \int_0^\xi \rho_*(f)df$*

Proof is in Appendix D.2. Eq equation 26 supports our insights drawn from Theorem 1, that the limiting risk should be proportional to the theoretical value of AU; Besides, equation 27 implies that it is reasonable to assume $\xi$ is close to 1 while the upper limit of $\varepsilon(t)$ is close to 0 as we state in Theorem 1 and Corollary 1.

### 3.2 BEYOND EQUILIBRIUM OF SMD

We have presented a detailed analysis on the evolution of the NRQ in equilibrium of SMD, showing that the convergence rate and limiting risk are mainly controlled by AU. Based on the insights drawn from the equilibrium case, we can infer how evolution of AU dominates the evolution of NRQ beyond equilibrium.

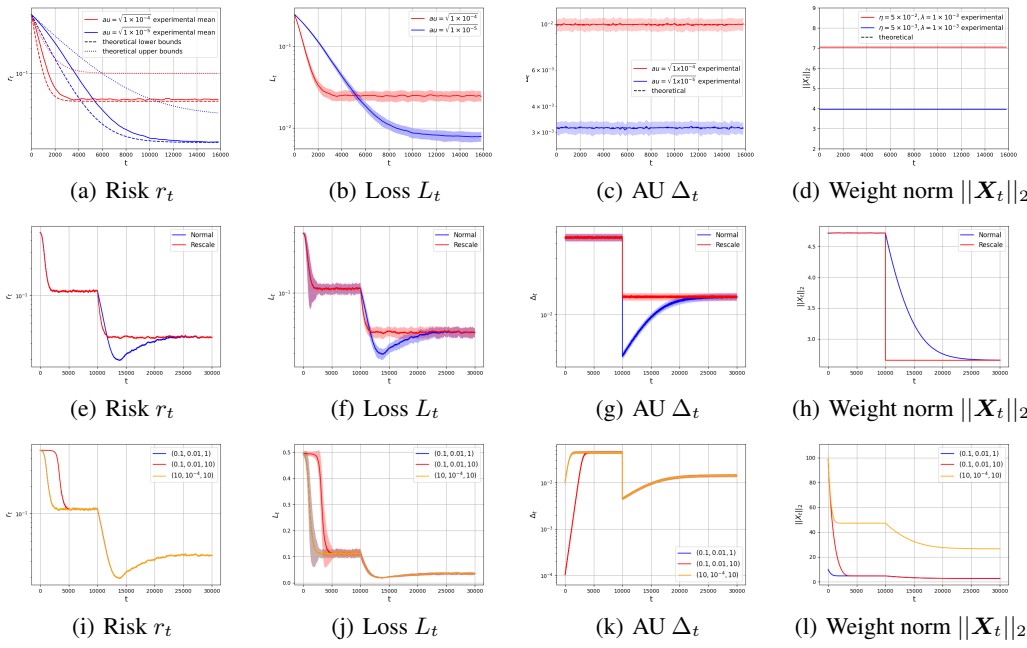

Figure 2: Experiments of NRQ. We exhibit the averaged results of 100 trials. (a)-(d): Evolution of NRQ in equilibrium state; (e)-(h): Escape behavior of NRQ after LR decay. LR is divided by 10 when $t = 10^4$, "Rescale" means $\boldsymbol{X}_t$ is divided by $(10)^{1/4}$ when learning is shrunk; (i)-(l): Evolution of NRQ with different and LR, WD factor and initialized standard deviation, denoted by $(\eta, \lambda, \tilde{\sigma})$. Blue lines are masked by yellow lines in (i), (j), (k) since they have exactly same evolution.

**"Escape" by the autonomous increase of AU** The following corollary shows the evolution of $\Delta_t$ can lead to an "escape" behavior of optimization trajectory, resulting in a temporary "decreasing, then increasing" risk:

**Corollary 2** (A sufficient condition of "escaping" behavior). *Given $\eta, \lambda$, if the following conditions hold: 1) $\exists \varepsilon > 0, \forall t > 0, \varepsilon(t) < \varepsilon < \underline{r}^*$; 2) $f_0 = 1 - \underline{r}^*$; 3) $M_0 > \frac{(p-1)\sigma^2 \eta}{(\underline{r}^* - \varepsilon)(a_2 - a_1)\xi}$. Then $\exists T > 0$,*

$$r_T < r_0 \leq \varliminf_{t \to \infty} r_t. \tag{28}$$

Proof is in Appendix E. equation 28 indicates a kind of "escape" behavior, because it implies that the trajectory of $\tilde{\boldsymbol{X}}_t$ will first approach an optimal point $\tilde{\boldsymbol{X}}^*$ at the beginning, and then depart from $\tilde{\boldsymbol{X}}^*$. Intuitively, this "escape" behavior is caused by increasing AU (which is also the main idea of the proof for Corollary 2): the initial weight norm $\sqrt{M_0}$ is sufficiently large, so $\Delta_t$ is relatively smaller than $\sqrt{2\lambda\eta}$ at the beginning, which allows the risk $r_t$ to reduce below the inferior of limiting risk given $\sqrt{2\lambda\eta}$ for a while. But when equilibrium is reached and AU increases to $\sqrt{2\lambda\eta}$, it will force the risk to go back to its limit value. See the blue lines in Figure 2 (e)-(h).

Even though Corollary 2 only gives a sufficient condition for the "escape" behavior in NRQ, such "escape" phenomenon can be seen in real data experiments in practice. Wan et al. (2021) exhibits a so-called "pseudo overfitting" phenomenon observed in CIFAR10 experiments with commonly used settings. They speculate "pseudo overfitting" is caused by temporarily increasing AU after LR decay based on empirical observations. Corollary 2 offers strong theoretical evidence for their conjecture, showing increasing AU indeed can lead to "escape" behavior under specific conditions. We also apply "rescale" strategy proposed in Wan et al. (2021) on NRQ, in which when LR is divided by $k$, weight norm is divided by $k^{1/4}$. The "rescale" strategy can indeed eliminate the "escape" behavior (see Figure 2, (e)-(h), red lines).

**Initialized value of weight norm** The evolution of AU can provide new interpretations on how initialization strategy affect the training of normalized model.

The weights of neural network are usually initialized as Gaussian $\mathcal{N}(0, \tilde{\sigma}^2 \boldsymbol{I})$, where $\tilde{\sigma}$ need to be carefully tuned. In mean field theory and NTK theory, standard derivation of Gaussian initializing strategy is crucial in obtaining good performance. Experiments on real datasets seem to support these theorems, where Gaussian initializing strategies with carefully tuned $\tilde{\sigma}$ such as Kaiming He et al. (2015) or Xavier Glorot & Bengio (2010) indeed outperform the naive Gaussian initializing strategy. However, when initializing normalized model, $\tilde{\sigma}$ only influences the initialized value of weight norm according to the large number theorem: $||\boldsymbol{X}_0||_2^2 = \sum_{i=0}^{p} (\boldsymbol{X}_t^{(i)})^2 \approx p\tilde{\sigma}^2$. The following corollary implies $\tilde{\sigma}$ are not so crucial for normalized model:

**Corollary 3.** $\forall k > 0$, if $\boldsymbol{X}_0$ is multiplied by $k$, enlarge $\eta$, $\lambda$ by $k^2$, $\frac{1}{k^2}$ times respectively, $r_t$ remains unchanged.

Proof is in Appendix E.2. Corollary 3 shows no matter how to set $\tilde{\sigma}$, as long as LR $\eta$ and WD factor $\lambda$ are adjusted accordingly, the evolution dynamics of NRQ does not change at all. Because the adjustment in Corollary 3 can remain the evolution equation 10 by maintaining the evolution of $\Delta_t$ (Comparing blue and yellow lines in Figure 2, (i)-(l)).

Furthermore, combining equation 17 and equation 11, we can interpret why initialization affect the evolution of NRQ: when $\lambda, \eta$ are given, the initialized value $M_0$ can change the evolution of AU $\Delta_t$ by changing equation 17, resulting in different training curve at beginning. But their limiting risk remains unchanged, since the theoretical value of AU does not change (Comparing blue and red lines in Figure 2 (i)-(l)); same phenomenon occurs when $M_0, \lambda\eta$ are fixed, but $\lambda, \eta$ change.

Though in NRQ, the conclusion that "same limiting AU will lead to similar limiting risk" is true, same conclusion does not necessarily hold on real data experiments. The two local minima of Rayleigh Quotient have exactly same geometric characteristics, but in real data experiments, the loss landscape may have multiple local minima with different geometric characteristics. Even though the theoretical value of AU is fixed, different evolution of aus may make the optimization trajectory get close to different local minima, resulting in different final performance. This is the reason why in practice, with fixed LR, WD factor, and $\tilde{\sigma}$ in Gaussian initialization still may influence the performance of neural network.

## 4 REAL DATA EXPERIMENTS

Aside from NRQ experiments, we also conduct experiments on CIFAR10 (Krizhevsky et al., 2009), and ImageNet (Deng et al., 2009) respectively to verify the insights drawn from NRQ. We use pure SGD without momentum in all real data experiments to eliminate the possible effect of momentum, though Wan et al. (2021) claim SGD with momentum has similar SMD mechanism. In CIFAR10 experiments, we adopt Resnet18 (He et al., 2015) as the baseline model; total epochs is 200; LR is divided by 5 at epoch 60, 120, 160; Batch size is 256. In ImageNet1000 (Deng et al., 2009) experiments, most settings follow Goyal et al. (2017), except LR is initialized as 1, momentum is 0. Smith et al. (2020) shows using pure SGD with larger LR can obtain similar performance as the standard SGDM setting.

Our experiments' results (Figure 3,4) show the insights from NRQ also hold in real data experiments: in cifar10 experiments, when $\eta\lambda$ is fixed, AU in equilibrium of SMD remains unchanged, so do the steady loss and Accuracy (Figure 3 (a)-(d)). But different LR and WD factor can affect the evolution at beginning. Wan et al. (2021) shows similar observations in ImageNet experiment; Figure 3 (e)-(h) exhibit the "pseudo overfitting" phenomenon shortly after the first LR decay (epoch 60). Rescaling strategy can avoid such "pseudo overfitting" by eliminating the increasing AU phenomenon after LR decay; From Figure 3 (i)-(l), when initialized weight is enlarged by 10, the AU is smaller at the beginning, hence the training loss (test accuracy) decreases slower (increases quicker). When the LR and WD factor are adjusted according to corollary 3, the evolution of AU remains unchanged, and so do the evolution of training loss/test accuracy. ImageNet experiments have similar phenomenon (Figure 4 (a)-(d)).

## 5 CONCLUSION

In this paper, we propose a simple yet representative normalized quadratic model, named as Noisy Rayleigh Quotient (NRQ), to study the effect of SMD on the evolution of SGD with WD. Our

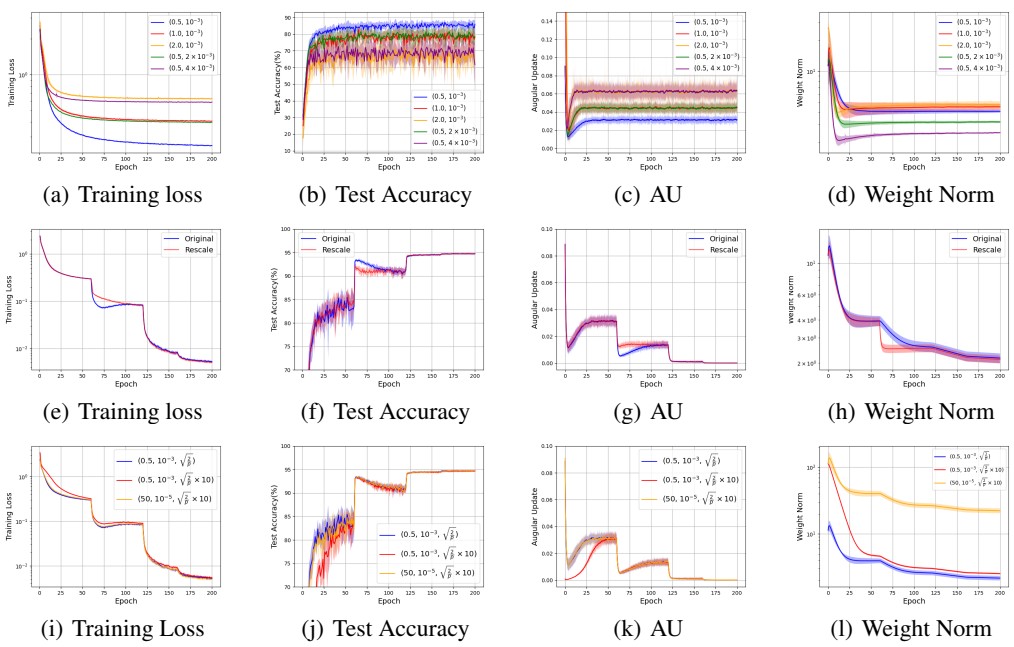

(a) Training loss     (b) Test Accuracy     (c) AU     (d) Weight Norm

(e) Training loss     (f) Test Accuracy     (g) AU     (h) Weight Norm

(i) Training Loss     (j) Test Accuracy     (k) AU     (l) Weight Norm

Figure 3: Resnet18 on CIFAR10, we exhibit the averaged results of 10 trials. (a)-(d): Training curves with different LR and WD factor, denoted as $(\eta, \lambda)$; (e)-(h): pseudo overfitting in CIFAR10 experiments. LR is 0.5, WD factor is $10^{-3}$, "rescale" means all weights is divided by $5^{1/4}$ at epoch 60; (i)-(l): Training with different LR, WD factors, and initialized standard deviation in convolution layer, denoted by $(\eta, \lambda, \tilde{\sigma})$. $\sqrt{\frac{2}{p}}$ denotes Kaiming Init (He et al., 2015).

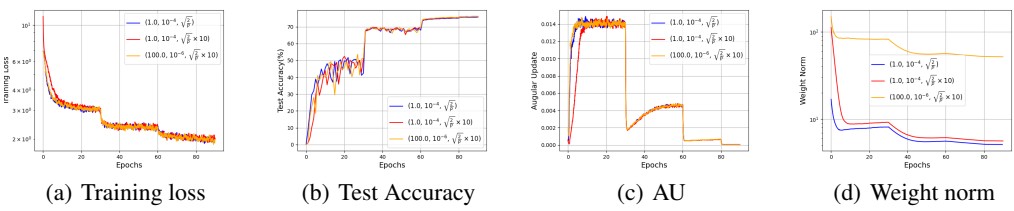

(a) Training loss     (b) Test Accuracy     (c) AU     (d) Weight norm

Figure 4: Resnet50 on Imagenet Training with different LR, WD factors, and initialized standard deviation in convolution layer, denoted by $(\eta, \lambda, \tilde{\sigma})$. $\sqrt{\frac{2}{p}}$ denotes Kaiming Initialization (He et al., 2015).

theoretical results show SMD influences the evolution of SGD by controlling AU, and AU can dominate the convergence rate as well as limiting risk of NRQ. Our real data experiments show the insights drawn from NRQ are consistent with empirical observations. We believe our theorems can deepen our understandings on the underlying mechanism of deep neural networks.

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

# A    APPENDIX

You may include other additional sections here.

