# OpenReview forum: "How Normalization and Weight Decay Can Affect SGD? Insights from a Simple Normalized Model"
_ICLR.cc/2023/Conference — Submitted to ICLR 2023_

### Official Review · Reviewer_UaUq · 2022-10-12

**Confidence:** 2
**Correctness:** 4
**Technical Novelty And Significance:** 2
**Empirical Novelty And Significance:** Not applicable
**Recommendation:** 5

**Clarity, Quality, Novelty And Reproducibility:**

The paper is generally well-written, but can be somewhat inaccessible to readers outside this particle niche of theoretical ML. The paper relies strongly on concepts from SDEs, e.g. Ito’s lemma. While giving a complete tutorial on SDEs is outside the scope of the paper, perhaps a little bit more background would make the paper more widely accessible.

While SMD has been proposed previously, the paper makes good advancements in analyzing models where this phenomenon occurs naturally. I would say that the novelty is about average.

The reproducibility is likely most relevant for the empirical part. The authors clearly state the hyperparameters and use many seeds to provide error bars, so the paper appears to be very reproducible.

Overall, I would encourage the authors to state the assumptions more clearer. E.g. The model given in eq 9 is stated as an approximation. It would be good to explicitly state the assumptions that go into this approximation. Furthermore, please state the assumptions when stating the theorems. The use of << in corollary 1 is also unclear, what does it mean?




**Strength And Weaknesses:**

Strengths:

- Normalized models are an interesting and relevant topic for theoretical work.
- The paper is well written.
- The empirical verification is throughout with e.g. many samples used to obtain error bars.

Weaknesses:
- The assumptions of diagonal A and diagonal and constant noise are rather strong.
- The paper is probably somewhat inaccessible to the wider ICLR audience.



**Summary Of The Paper:**

Deep neural networks typically contain normalization layers like layer norm or batch norm, which causes scale invariance in the network parametrization. This paper studies learning dynamics in this setting. Firstly, the authors introduce a simple optimization problem dubbed Noisy Rayleigh Quotient (NRQ) which exhibits this scale-invariant property. The authors then analyze the optimization behavior of this model under an SDE approximation for SGD. Under diagonal assumptions and constant noise, the authors give analytical expressions for the dynamics and limiting distribution of NRQ. At last, the authors verify that their theoretical results match experimental data.



**Summary Of The Review:**

The authors introduce a natural model (NRQ) for studying the optimization of a normalized model and give theoretical results under strong but natural assumptions. The paper clearly shows how the theoretical results relate to empirical work, but the language and narrow topic make the paper somewhat inaccesible to a wider audience.

---

### Official Review · Reviewer_ucmd · 2022-10-24

**Confidence:** 4
**Correctness:** 2
**Technical Novelty And Significance:** 3
**Empirical Novelty And Significance:** 3
**Recommendation:** 5

**Clarity, Quality, Novelty And Reproducibility:**

See my comments on weakness. The presentation can be improved a lot. To judge about novelty, authors need to review the literature on Oja's flow and stochastic power methods.

**Strength And Weaknesses:**

**Strength**

Studying Oja's flow on leading eigenvalue problems is an important topic in stochastic approximation. Despite recent progress, there are many open-problems in stochastic power iteration methods. This paper studies a continuous-time model of Oja's flow that may enable simplifying the complexity of analyzing Oja's flow.

**Weakness**
- *Motivation.* In my opinion, there is a gap between optimization with batch normalization in neural nets and the eigenvalue problem. It is better to formulate the problem in a standard theoretical context, such as stochastic power iteration. This setting is standard and well-studied in the literature, which allows authors to compare.
- *Restrictive assumptions.* Assumptions are rather strong and are not validated or discussed in the paper.
  - Lemma 1 assumes that a random process is constant over time.
  - Assumption 1 enforces the noise of stochastic gradient is isotropic.
  - Assumption 2 limits the analysis to find the leading eigenvector of a diagonal matrix.
- *Generalization.* For a very limited toy example of an eigenvalue problem, the paper extrapolates their findings on *the escape phenomenon* to the setting of learning residual networks with 18 layers for image classification. I found this unnecessary. In my opinion, the settings of stochastic power methods are well-established.
- *Soundness.* I am not sure about some of the statements in the paper.
  - Equation 11 is a random process (due $P_t$) while it is assumed to be deterministic in the integration of Eq. 12.
  - Why does corollary 1. not need the same assumption as Lemma 1?
- *Presentation.*
  - Notation $\epsilon(t)$ used in Thm. 1, Corollary 1 is not defined.
  - Would be great to define escape phenomena mathematically and then characterize the required time for escape. I did not grasp the notion of escape.




**Summary Of The Paper:**

This paper studies diffusion model of Oja's algorithm for leading eigenvector problem. When the noise is Gaussian and isotropic, the matrix is diagonal, then author establish diffusion convergence to equilibrium state. They show that the algorithm exhibits a escaping property in they laboratory setting which is similar to SGD dynamics on neural networks with batch normalization.

**Summary Of The Review:**

I recommend authors present their theoretical results in context of stochastic power iteration after removing the restrictive assumptions. Going from power iteration with restrictive assumptions to the standard settings of training neural nets is not sound and convincing.

---

### Official Review · Reviewer_mwkM · 2022-10-24

**Confidence:** 3
**Correctness:** 3
**Technical Novelty And Significance:** 3
**Empirical Novelty And Significance:** 3
**Recommendation:** 5

**Clarity, Quality, Novelty And Reproducibility:**

As regards the novelty, I believe that the contribution is probably sufficient, but the clarity of the paper is my main concern. In my opinion, the potential impact is limited by the writing of the paper which is a bit confusing and non-accessible, while the intuition is missing in most places.

**Strength And Weaknesses:**

I think that understanding the optimization dynamics for any model class is a very interesting problem. However, in my opinion the actual problem that the authors aim to analyze is not clearly defined and is not motivated well in the current version of the paper. As a non-expert in the field I believe that the related work is properly cited, but on the other hand I find the whole story of the paper a bit incoherent. The authors build on previous work that is assumed to be known, which I think makes the paper rather confusing and non accessible. The technical content seems to be correct, but I have not checked it in details. Also, a lot of simplifications and assumptions are used such that to develop the theoretical results, which might be a bit too optimistic/unrealistic compared to the actual scenarios.

**Summary Of The Paper:**

The authors propose a theoretical analysis for the Stochastic Gradient Descent optimizer for normalized deep neural networks including weight decay. In particular, they study the behavior of the loss during training, taking into account that for such models the optimization converges to an equilibrium related to the weights norm. For the theoretical analysis a simple optimization problem is used. Empirical demonstrations verify the analysis.

**Summary Of The Review:**

Questions:
1. In my opinion some basic concepts should be explained more thoroughly. For example, you can use a 2-dim optimization problem (Eq. 1) together with an appropriate figure, in order to explain clearly what is the Spherical Motion Dynamics (SMD), what is the Angular Update (AU), what is the equilibrium state, etc. In general, I believe that the paper should be standalone.
2. The scale-invariant model is the same as the normalized model? Can you give some examples for each of them.
3. Let's consider a 2-dim optimization problem (Eq. 1). The optimization trajectory of a scale-invariant model (under SGD) behaves as a motion on the sphere. Following your example, let assume that the minimizer is the direction $e_1=\pm [1,0]$. When the optimizer hits this valley of minima, then the sphere motion implies that the optimizer stays in this valey following an up-down trajectory?
4. When we include the weight decay in the problem above, then the SMD applies. What is the implication? When the optimizer hits the valley of minima, then the trajectory moves closer to 0? And what the "equilibrium" state means? Does it mean that first the correct direction is found, then the norm of the weights is minimized (close to 0 for your problem), and then the optimizer due to the noise moves up-down?
5. In Sec 2.1 a deterministic optimization problem is proposed together with a stochastic optimization procedure. I wonder what is the purpose of Remark 1? Does it imply that NRQ is harder than NQM? I think that NRQ without the unit norm constraint is not an equivalent problem.
6. In Sec. 2.2 the paragraph before Eq. 10 is a bit confusing. Also, why Eq. 9 can be rewritten as Eq. 10 - 11 and get the associated solution?
7. The sections 2 and 3 are quite confusing and unclear as well. In most cases the intuition is missing, there are a lot of technical derivations with no clear goals, and in general I think that the presentation is not coherent and accessible.

In general, the problem that the paper aims to study is very interesting, and perhaps the associated technical content. However, I think that the current version of the manuscript probably does not reflect its actual contribution and merits. In my opinion an improvement mainly on the clarity and the accessibility of the paper is necessary.

---

### Official Review · Reviewer_y88R · 2022-10-30

**Confidence:** 4
**Correctness:** 3
**Technical Novelty And Significance:** 2
**Empirical Novelty And Significance:** 2
**Recommendation:** 5

**Clarity, Quality, Novelty And Reproducibility:**

The writing is overall clear, yet still has room for improvement. The convergence analysis of NRQ model is novel.  Overall the theorem statement and lemmas look reasonable to me, though I didn't check all the details in the proof.

**Strength And Weaknesses:**

##Pros:

1. Scale invariance in modern deep netowrks has been an hot and important topic in deep learning theory, due to the significance of various normalization layers in optimization. However, the community lack of good understanding of the training behavior of scale invariant models under SGD. Though there exists some theoretical analysis and hypothesis based experiments for general deep learning loss, it is not clear how the dynamics look like, e.g. the convergence rate of training loss, even for any simple model, because scale invariance also changes the loss landscape drastically, making it inherently non-convex.  This paper provides the first convergence analysis for concrete scale invariant loss functions, that is, on the NRQ model.

2. Though anguler update (AU) is implicitly defined in the previous papers, this paper gives the first mathematical rigorous defition of AU and prove that it is equal to the $\sqrt{2\eta\lambda}$.

3. This paper also makes some analysis or intuitions from previous works more concrete, at least in the setting of NRQ. For example, as pointed out in [Li et al.,20], when decreasing learnign rate $\eta$, the training loss can first decrease and then increase, due to the change of the parameter norm.


##Cons:

1. Though the theoretical convergence result is solid, but the interpretation and some phrasing are confusing. The paper is written in the way that AU is the cause of various unconventional optimization behavior of the scale invariant models. But in fact, AU in this paper, espceially under the assumption that trace of covariance is constant on unit norm ball, it merely a function of LR $\eta$, WD $\lambda$ and time $t$. In the end of page 6, the authors wrote that "showing that the convergence rate and limiting risk are mainly controlled by AU". I do not see any casual relationship between AU and convergence rate or limiting risk. They are all just consequences of SGD or SDE on a scale invariant loss. Thinking convergence rate or limiting risk as functions of AU doesn't seem to bring any conceptual convenience, nor convenience in the mathematical proof. For example in the proof Theorem C.0.3 in appendix, AU is not used at all. Even for the main theorem (Theorem 1), which has AU in its statement, it only follows from Theorem C.0.3 with some substitution, namely replacing the parameter norms by AU, which are equivalent up to simple transformation. Given that such equivlance relies on the assumption that trace of corvariance is constant exactly, it is not easy for me to see the benefit of explaining convergence in terms AU.

2. Corolary 3 is already known. The SGD case is proved by [Lemma 2.4, Li et al.,19]  and the SDE case follows directly from Eq. (9) and (10) in [Li et al.,20]). I will not treat Corolary 3 as a contribution. Moreover, the derivation (even in this paper) does not use the concept of AU at all.

3. The authors mentioned that in the abstract "theoretically prove SMD can dominate the whole training process via controlling the evolution of angular update (AU), an essential feature of SMD". It's not clear what "SMD can dominate the whole process mean". If it means SDE is a good approximation like we can ignore the impact of full-batch gradient on norm, then I cannot find the proof in the current draft. Also, AU is not a feature of SMD, as it can be defined for any dyanmics. The feature of SMD is that AU converges to a time-independent value under some assumptions.

4. Missing related work: [Wang and wang, 2022] also analyzes convergence rate and gives a more fine-grained characterization for the distribution of the direction of the parameter under more general settings -- their analysis holds for all morse function, including NRQ.

##Minor commens:**

1. Figure 1. missing bracket, should be $(x^2+y^2)/(x^2+y^2) +(x^2+y^2)$.
2. In equation (5), no bracket is needed around $\tilde \Sigma$.
3. On top of page 4, "Eq equation 11" appeared twice.
4. In defition 1, it seems "if" is missing.
5. In equation (20), $\tilde G_\tau$ is undefined.
6. I found there is quite a gap between Theorem 1 and Theorem C.0.3. Might worth elaborate the proofs here.
7. On page 7, "temporarily increasing AU", should be "temporarily decreasing AU"?

##References:

- Li, Zhiyuan, Kaifeng Lyu, and Sanjeev Arora. "Reconciling modern deep learning with traditional optimization analyses: The intrinsic learning rate." Advances in Neural Information Processing Systems 33 (2020): 14544-14555.

- Li, Zhiyuan, and Sanjeev Arora. "An exponential learning rate schedule for deep learning." arXiv preprint arXiv:1910.07454 (2019).

- Wang, Yi, and Zhiren Wang. "Three-stage Evolution and Fast Equilibrium for SGD with Non-degerate Critical Points." In International Conference on Machine Learning, pp. 23092-23113. PMLR, 2022.

**Summary Of The Paper:**

This paper studies the dynamics of SGD with WD on a particular scale invariant loss function, the so-called Noisy Rayleigh Quotient (NRQ). The main contribution of the paper is the convergence analysis for NRQ under continuous approximation.


**Summary Of The Review:**

Though the convergence analysis on NRQ model is a solid contribution, I think the current story this paper tries to deliver could be misleading to the community. Besides, the interpretations are more or less already known from previous works. Thus I think this paper does not meet the bar of ICLR.

---

### Decision · Program_Chairs · 2023-01-20

**Decision:**

Reject

**Justification For Why Not Higher Score:**

N/A

**Justification For Why Not Lower Score:**

N/A

**Metareview: Summary, Strengths And Weaknesses:**

This paper studies how normalization and weight decay can affect SGD through a simple normalized model. The reviewers raised many good questions, but the authors did not respond. Therefore, I have to recommend rejection.

**Summary Of Ac-Reviewer Meeting:**

N/A